# City Regeneration through Modular Phase Change Materials (PCM) Envelopes for Climate Neutral Buildings

**Javier Orozco-Messana** [1,*], **Vicente Lopez-Mateu** [2] and **Teresa M. Pellicer** [3]

1   Institute of Materials Technology (ITM), Universitat Politécnica de Valencia, Camino de Vera s/n, 46022 Valencia, Spain

2   Departamento Construcciones Arquitectónicas, Universitat Politécnica de Valencia, 46022 Valencia, Spain; viloma@upv.es

3   Departamento Ingeniería de la Construcción, Universitat Politécnica de Valencia, 46022 Valencia, Spain; tpa@upv.es

*   Correspondence: jaormes@cst.upv.es

**Abstract:** Climate change is driving urban development policies for nearly all cities, which are responsible for over 40% carbon emissions in the world. UN SDG 11 ("Make cities and human settlements inclusive, safe, resilient and sustainable") defines critical indicators focused on carbon footprint reduction through green policies and city heritage preservation. Urban regeneration should ensure climate comfort for citizens while enhancing legacy urban resilience. New solutions for urban regeneration such as Phase Change Materials (PCMs) provide inexpensive energy adaption solutions by reducing peak thermal loads, and their market share is growing yearly by 16% (OECD market trends). However, these materials must be integrated into recyclable flexible building elements to ensure tailored responses to different seasons and climates. Modular PCM elements working together with Passive Haus techniques have demonstrated their flexibility. This paper presents a new, efficient, and sustainable modular solution for PCM-based building envelope regeneration projects implemented jointly with Passive Haus strategies and Nature-Based Solutions (NBS) at street level. The efficiency of the proposed strategy is demonstrated though a simplified Digital Twin of the Benicalap neighbourhood in Valencia, Spain. The model simulates the climate evolution at the neighbourhood level, and can be used in any urban background to obtain a new carbon footprint which is then used as the main criterion for joint impact assessment of the proposed modular PCM-based building envelopes.

**Keywords:** neighbourhood sustainability; PCM; Passive Haus; NBS; sustainability assessment

## 1. Introduction

City regeneration is a key driver of sustainable urban performance. Urban adaption is the evolutionary process through which city life is applying regeneration policies in order to meet a set of formal (and informal) objectives. Nowadays, SDG 11 develops the widest consensus on sustainability targets for the regeneration of urban environments by setting Key Performance Indicators (KPIs) to evaluate how cities are performing on sustainable growth [1]. The guiding principles for sustainable and resilient urban expansion combine requirements in terms of social habits, services, and processes within an economic context. Cities are living organisms with a metabolism controlled by urban policy, mostly implemented through regeneration schemes, wherein urban legacy systems (heritage) have dominant relevance [2]. Carbon efficient climate adaption is the common denominator in the ever-increasing problems faced by cities, including social disintegration, economic recession, environmental pollution, and deterioration of urban function. New modular PCM-based solutions for urban renewal are leading the renovation wave worldwide [3,4] by shifting climate thermal loads to the benefit of building energy efficiency; however, they fail to provide an efficient and sustainable solution.

The combined impact of climate change and urban growth under local, regional, and global constraints are increasing urban energy needs and greenhouse gas emissions exponentially. The associated carbon footprint of built-up environments is increasing yearly beyond the current 40% share of the world mark, and should be addressed through buildings' thermal performance within their own context [5]. The combined environmental impact of urban heat islands and extreme climate events can be best addressed through a hybrid strategy for optimal building thermal performance [6]. However, impact assessments of urban regeneration strategies do not consider joint evaluation.

The balanced contribution of Nature-Based Solutions (NBSs), Passive Haus design strategies and modular (circular) building envelope solutions have not been properly addressed [7]. The dominant research networks can be seen in Figure 1, obtained from a full bibliography search performed using the Dimensions app [8] and the topics PCM, NBS, and Passive Haus. In total, 14,581 published papers (limited to the period 2016–2022) were considered, with a mean citation index of 14′83; however, all of them failed to address the previously identified research gaps.

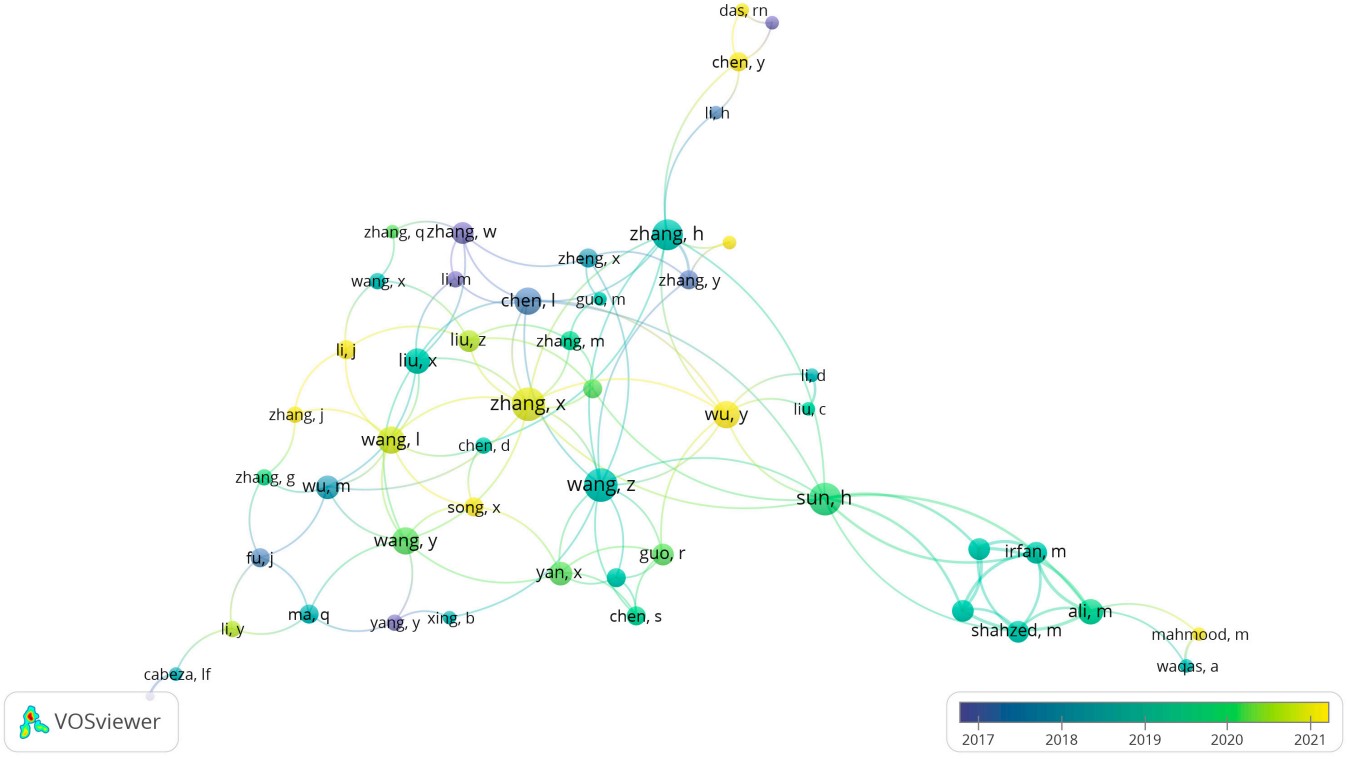

**Figure 1.** VOS diagram of bibliography network based on PCMs, NBS, and Passive Haus.

This paper seeks a new ceramic PCM-embedded sandwich-type modular design for building envelope regeneration and develops a combined strategy for an efficient contribution to the challenge.

## 2. Background Literature Review

The research objective for this paper was set based on establishing a new efficient design for a modular panel to be used in the regeneration of existing building envelopes implemented in combination with Passive Haus techniques and NBS support for enhanced neighbourhood impact minimizing the carbon footprint of residential buildings.

As can be seen in Figure 1 above, PCM research is evolving rapidly, developing more stable and sustainable alternatives [9]. New strategies provide higher active surfaces for facilitating heat exchange [10] and combined architectural solutions for buildings lead to enhanced results [11]. Market opportunities are growing exponentially due to the major social thrust for climate-neutral cities [12], shaping the research focus on this topic.

Starting from the previous bibliographical search and following leading authors according to their references, a detailed analysis of results contributing to the background of the proposed research was extracted. Relevant background topics were identified (see bullet points below) and the following conclusions established:

- Long term energy performance under different climate conditions has been well studied by Pajek et al. [13], including climate change scenarios; they concluded that a combination of tailored envelope designs with passive ventilation renders the best energy performance.
- Urban retrofitting policies have been reported as an emergent opportunity for fighting climate change [14] in most climate scenarios.
- Under dominant climate conditions, including climate change scenarios, the most efficient building refurbishment solutions relate to envelope interventions focused on materials capable of reducing the building's peak energy loads [15].
- In PCM module design, modularity is required for implementing circularity of the regeneration components (material carbon footprint reduction) as well as minimum installation and operation costs (easy maintenance). The building renovation wave should consider adequate criteria on materials reuse and incorporation into standardized modular elements, fitting with the regeneration market [16,17]. The main conclusion is that sandwich layers (insulation and/or PCM) held in place by standard size hard surfaces can meet user requirements.
- For selection of relevant commercially available PCM, several points are relevant and have been addressed by al-Yashiri et al. [18] in their analysis:
    - PCMs should be selected from cheap sustainable alternatives with high solidification enthalpy and cycle stability;
    - They require a conductive rigid support for best performance;
    - The PCM layer should be closest to the warm face of the application;
    - The melting temperature should be close to the comfort requirements.
- Following an extensive review and performance analysis of the main building envelope design solutions in different climate conditions, Arumugam et al. [19] concluded that:
    - Natural cross-ventilation both in internal and external faces is critical;
    - Humidity control through green infrastructure is desirable;
    - Insulation integration performs well in the most critical condition;
    - Leakage is critical and should be considered together with maintenance requirements and fire risk.
- External NBS influence: As demonstrated through the wide analysis of connected green infrastructure deployment at street level by Marando et al. [20], a street cover of 16% enhances the external temperature conditions by 1–2 °C while controlling wind, radiation, and humidity impacts.
- Internal heat load control (passive systems): Controlling internal heat loads is critical for assessing the relevant impact of PCM-based strategies. According to Coma et al. [21], thermal loads should be distributed and balanced through ventilation and vegetation-based humidity control. According to Moussavi et al. [22], a reduction of 1–3 °C in internal temperature can be expected.
- A comprehensive approach to impact assessment evaluation of the proposed combined strategy for passive climate mitigation through PCMs relies on a full Life Cycle Assessment (LCA) at the neighbourhood level through the simulation of expected physical and weather conditions [23].

## 3. Materials and Methods

### 3.1. PCM Modules

State-of-the-art PCM based sandwiches [18] lack high thermal exchange surfaces. The proposed PCM sandwich was fabricated over a micro-structured stoneware scaffold

covered with a thin graphite film, which provides a thermal exchange surface several orders of magnitude higher than current solutions.

The scaffold was fabricated as an inverse replica of a 2 μm (average pore size) polyurethane foam soaked with a 60% solid content Standard Stoneware slurry (Euroatomizado), then burned to 300 °C (with reducing flame) to eliminate the polymer, leaving an extremely thin (conductive) black smoke layer for thermal conductivity enhancement. The obtained microstructure can be seen in Figure 2.

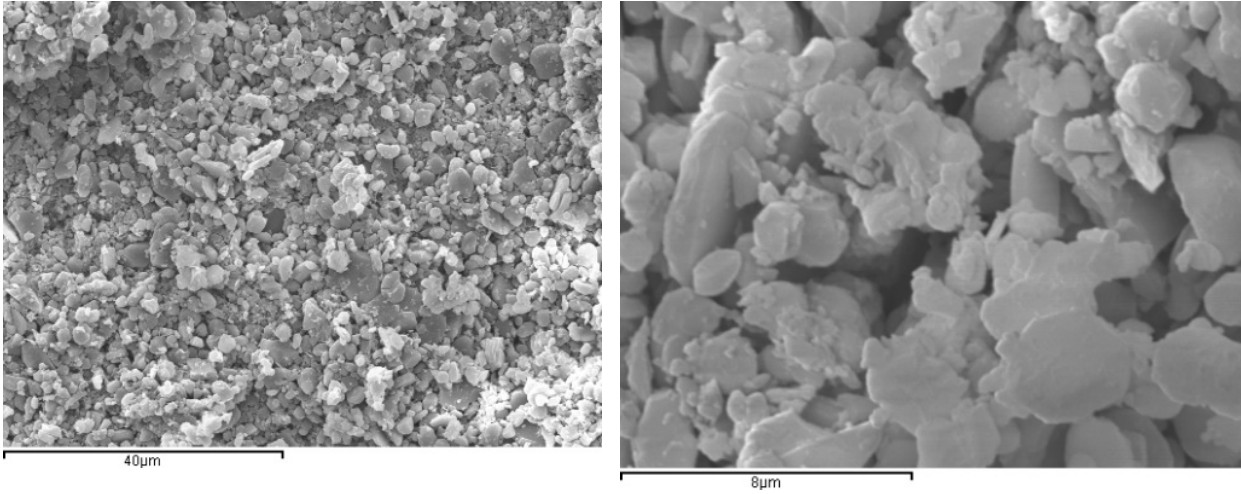

**Figure 2.** PCM ceramic scaffold after green compaction with black smoke film.

The flat tiles (1000 × 1000 mm with the desired thickness) of the scaffold were then sintered to 1000 °C in an oxygen-free atmosphere, converting the black smoke layer to a thin pyrolytic carbon film fully covering the whole internal open-pore structure with a dense and thermally conductive network.

After obtaining the ceramic scaffold, its open pore structure must be vacuum infiltrated with the selected PCM in its liquid phase.

The PCM materials used for finishing the sandwich were industrial-grade products [18]. The lot selected for the pilots (see Table 1) fulfilled the conditions of optimal performance in cold/warm weather, high phase change enthalpy, and a liquification temperature in the range of 20–25 °C range.

**Table 1.** PCM materials (from [18]).

| PCM Type | Reference | Manufacturer | T (°C) | Enthalpy (jul/gr) |
|---|---|---|---|---|
| Organic | RT21HC | Rubitherm | 21 | 150 |
| | PureTemp23 | PureTemp | 23 | 203 |
| | Beeswax25 | BASF | 25 | 142 |
| Inorganic | DS5040X | BASF | 24 | 168 |
| | MPCM24D | Microtek | 25 | 197 |

PCM panels one square meter in size were packed with an additional polyisocyanurate insulation layer of the desired thickness and then wrapped in aluminium foil for best conductivity and vapor barrier layer. The two external layers in the panel were placed with a 1 cm misalignment for easier assembly (see Figure 3).

The final sealing of the panel was prepared with a paste made from asphalt rubber incorporating reclaimed tire microchips (5% weight). This sealing material performs extremely well for over 50 years in the operational temperature range, and has a very low carbon footprint.

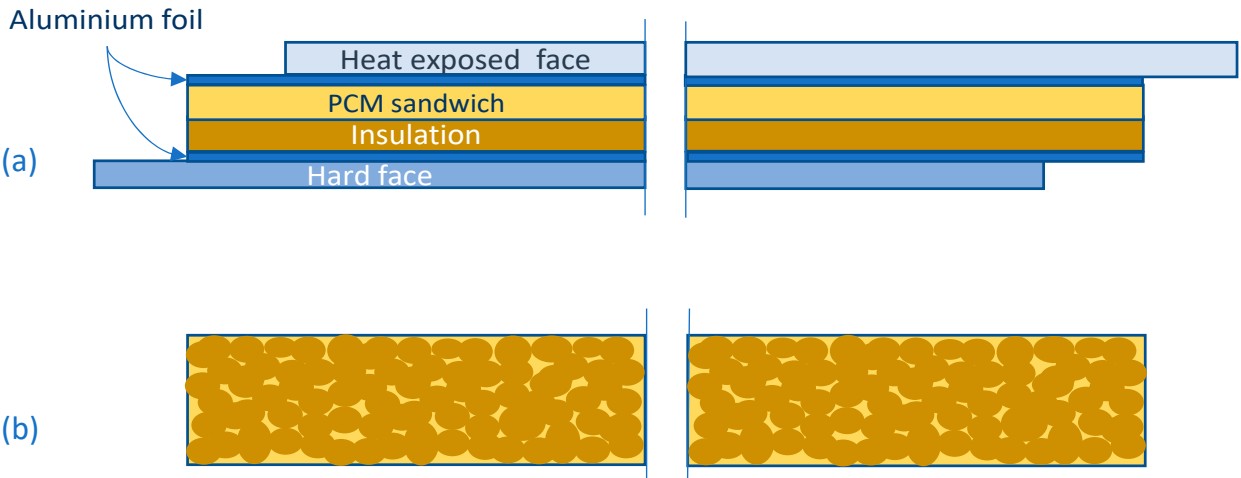

**Figure 3.** PCM panel design: (**a**) panel cross section and (**b**) PCM sandwich.

Sample panels were characterized in an accelerated performance evaluation with twenty cycles of 1 h each (see Figure 4). The first ten cycles corresponded to an extreme summer (50–15 °C cycles, 30 min at each temperature, with an internal temperature of 23 °C). The second lot of ten cycles followed extreme winter conditions (20–5 °C cycles, 30 min per temperature, again with an internal temperature of 23 °C).

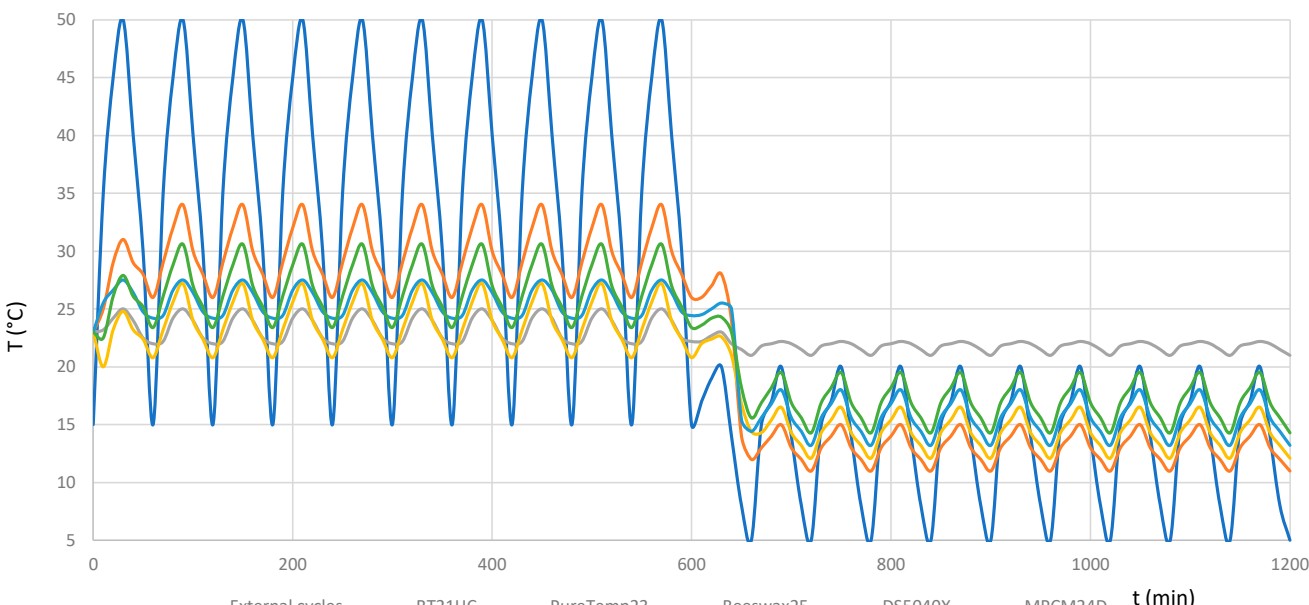

**Figure 4.** Accelerated Summer/Winter testing cycles.

The experimental setup is displayed in Figure 5. The external temperature was fixed in a closed water cycle while the internal temperature included air flow. Power savings were calculated for one complete PCM phase change cycle. After a careful assessment of the results, the PureTemp23 panel performed best as a panel (PCM producer PureTEMP, https://puretemp.com/?page_id=173 (accessed on 18 July 2022)), as can be seen from the experimental results presented in Table 2 using a standard wall with normal insulation and later placing the PCM sandwich on the correct side according to the season. Energy

savings corresponded to the average seasonal energy savings (summer and winter reference temperatures) from the season energy consumptions through the following ratio:

$$Energy\ saving = 100 \left\{ \frac{m_{PCM}\ LH_{PCM} + \max[abs(\Delta T)]\ \sum_{j=1}^{4} Ce_j\ m_j}{t_{cycle}\max[abs(\Delta T)]\ \sum_{i=1}^{4} \frac{e_i}{K_i}} \right\} \tag{1}$$

where $m_x$ is the mass of material "$x$", $LH_{PCM}$ corresponds to the latent heat of the PCM material, $Ce_i$ corresponds to specific heat of material "$i$", $e_i$ is the thickness of material "$i$", $K_i$ is the heat conductivity of material "$i$", and $t_{cycle}$ is the time (in seconds) for one measurement cycle.

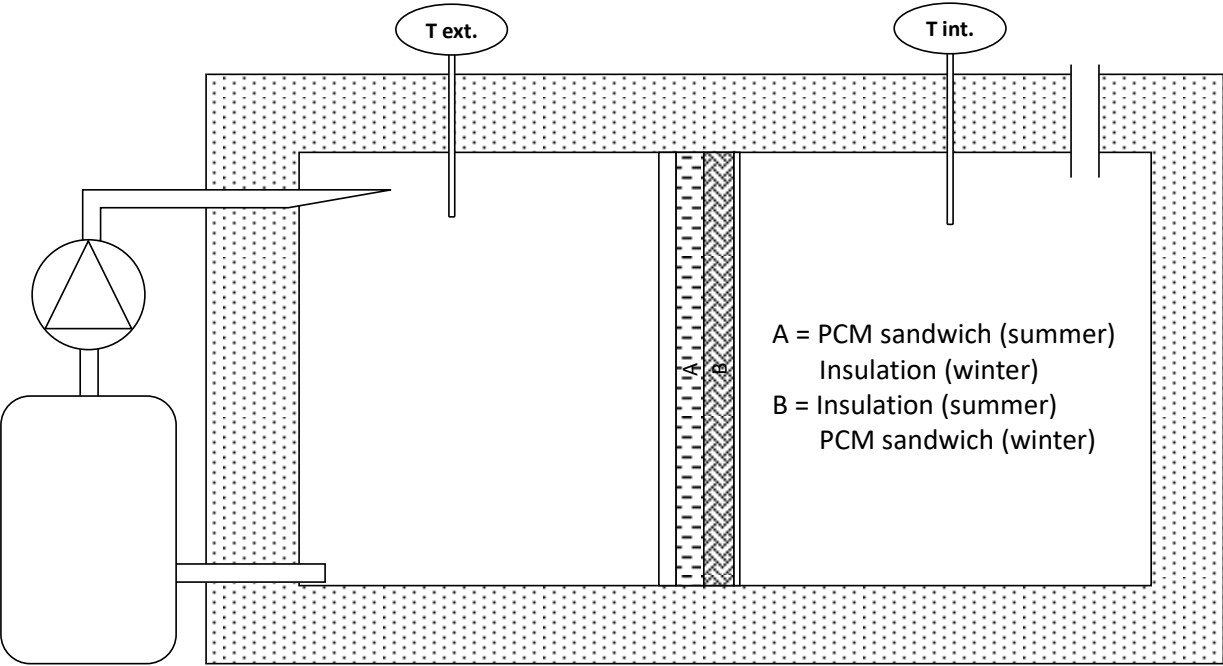

**Figure 5.** Experimental setup.

**Table 2.** PCM sandwich evaluation.

| Reference | Maximum $\Delta T$ (°C, Summer) | Maximum $\Delta T$ (°C, Winter) | Energy Saving (%) |
|---|---|---|---|
| RT21HC | 11 | −12 | 10 |
| PureTemp23 | 2.6 | −2 | 25 |
| Beeswax25 | 3.6 | −10.9 | 12 |
| DS5040X | 3.7 | −9.7 | 19 |
| MPCM24D | 7.6 | −8.6 | 17 |

### 3.2. Neighbourhood Model

For evaluating the relevance of the proposed PCM panel at the neighbourhood level combined with passive air flow in buildings and the implementation of NBSs, a digital model of a neighbourhood must be obtained. The proposed methodology is presented on Figure 6 according with the procedure developed by Orozco et al. [23,24]. The research was implemented in a peripheral neighbourhood of Valencia, Spain, called Benicalap, which has a hot semi-arid climate according to Köppen Climate Classification [25].

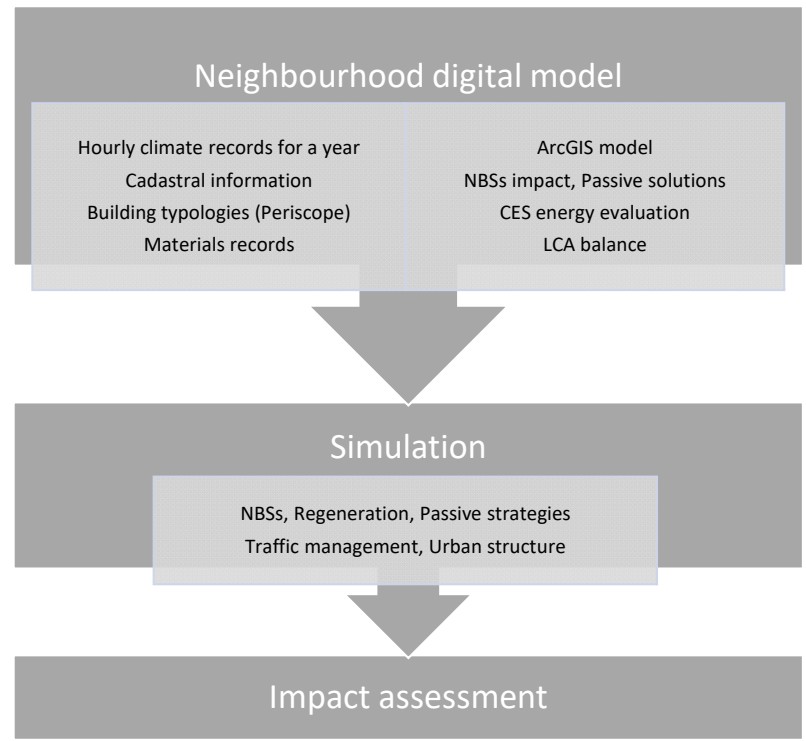

**Figure 6.** Procedure followed for developing the neighbourhood digital model [24].

The process described in Figure 6 was implemented using ArcGIS Pro (ESRI) software [26]. Each building in the digital neighbourhood model was matched to a building typology from the Tabula/Episcope [27] app, which includes all information related to materials and average energy use. Building typologies relate to the historical urban period and include specific materials used for each of the main subsystems (structure, envelopes, installations, and partitions) referenced from the Valencian Building Institute (IVE) [28] construction database. The summary of all materials (including thermal properties and carbon footprint) per building provides the embedded carbon footprint for materials and the energy footprint from yearly energy consumption, obtained using City Energy Analyst (CES) open-source software [29]. The process was repeated for all buildings in the model automatically through the ArcGIS Python app [30] routines, allowing calculation of Benicalap's carbon footprint. The Benicalap neighbourhood model [23] is shown on Figure 7.

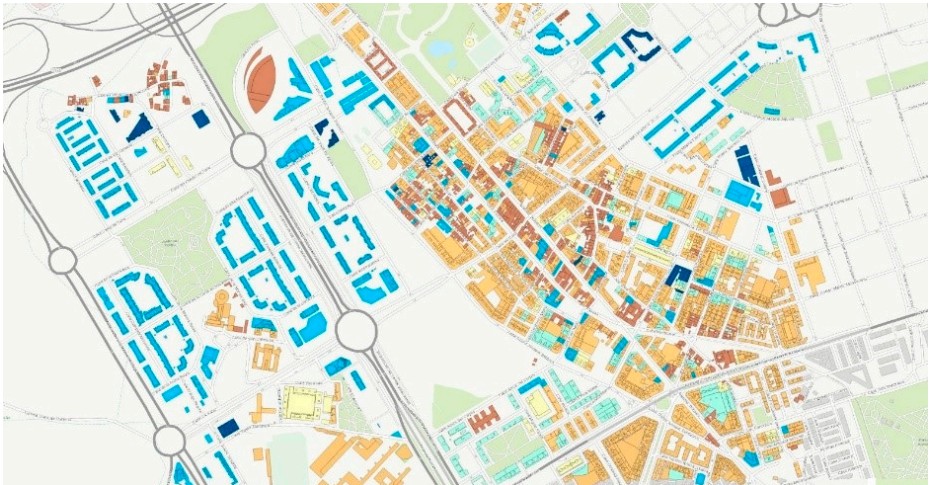

**Figure 7.** Benicalap Digital Twin.

Energy usage for climate systems in different buildings is calculated for a thermal load in a typical year (2019 in this case, avoiding any impacts from the COVID-19 pandemic), using 18 °C in winter and 23 °C during summer as a baseline comfortable temperature.

Energy calculations were performed for the evaluation of combined strategies on each independent building according to the bibliographic evidence presented in Section 2, taking into account the following assumptions:

- Street temperatures were taken from the Valencia Open Data portal [31], which corresponds to the climate station in Benicalap. Because the meteorological station for Benicalap is not within a green area, those streets with a green area over 16% area coverage are considered to have a temperature 2 °C lower.
- When a building is refurbished, PureTemp23 PCM panels will be installed and internal air flow ducts installed from the warm to the cold face, reducing the comfort temperature requirements by 2 °C.

The Digital Twin was then used again to evaluate the combined impact (green infrastructure and internal passive ventilation) of the PCM panel regeneration strategies on the neighbourhood carbon footprint independently for each building typology.

### 3.3. Methods and Proposed Research

This paper focuses on the combined evaluation of urban regeneration strategies based on the deployment of efficient PCM panels into a real neighbourhood. We obtained excellent results through the characterization of a new PCM panel (see Section 3.1) designed for enhanced efficiency.

The evidence used in this research was based on the LCA analysis (through the carbon footprint evaluation) of the proposed strategy and a further check of the economic viability of the alternatives. In the results, independent figures are obtained for each building typology by comparing the results prior to regeneration (actual LCA) and the independent simulations for each building typology in order to confirm the economically optimal outreach of the regeneration strategy.

Additional information was obtained from the Valencia Open Data portal [31] and stored in the digital model for LEED and BREEAM evaluations. This in turn offered additional dimensions for impact assessment evaluation of the regeneration strategies for Benicalap (Figure 8).

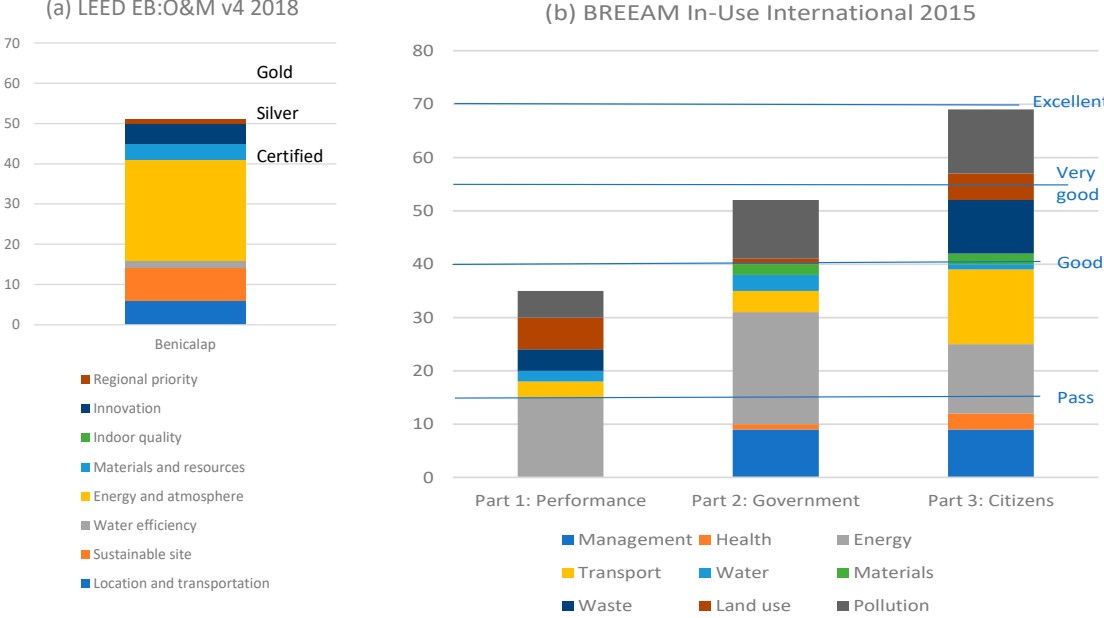

**Figure 8.** Evaluations of Benicalap neighbourhood from ISO 14040:2006.

## 4. Results

As was mentioned on Section 3.1, the first result is the new modular PCM panel design incorporating PureTemp23 PCM embedded in a stoneware foam structure. The characterization results show very efficient performance with excellent deployment projection and competitive costs.

The results obtained from the Benicalap Digital Twin related to LCA assessment were proven to present an average error of 10% [23]. The digital model provides both the actual (per building) carbon footprint estimation for a given climate record and a simulation of the same results based on different conditions. Following our research, the obtained results are presented in Table 3, which compares the actual calculations for the 2019 climate record to the projected results with complete refurbishing of all buildings per building typology, including PCM panel installation according to the technical requirements (i.e., insulation is only required for buildings constructed before 1979), and internal ventilation.

**Table 3.** Carbon footprints in Benicalap neighbourhood.

| Building Typology (Year Interval) | 2019 $CO_2$ Footprint (tn$CO_2$/Year) | | 2019 Refurbished $CO_2$ Footprint (tn$CO_2$/Year) | | Investment Return (Months) | Number of Buildings |
|---|---|---|---|---|---|---|
| | Embedded | Usage | Embedded | Usage | | |
| <1900 | 0.09 | 1.82 | 0.17 | 1.36 | 41 | 34 |
| 1901–1936 | 0.42 | 4.72 | 1.54 | 4.28 | 36 | 139 |
| 1937–1959 | 0.63 | 6.37 | 2.85 | 5.21 | 38 | 184 |
| 1960–1979 | 1.20 | 9.85 | 5.37 | 7.74 | 27 | 745 |
| 1980–2006 | 0.64 | 2.77 | 1.36 | 2.15 | 21 | 370 |
| 2007–2021 | 0.82 | 2.07 | 1.10 | 1.83 | 14 | 84 |
| TOTAL | 3.80 | 27.60 | 12.39 | 22.57 | 29 | 1556 |

In Table 3, the return on investment is calculated from the expected regeneration costs (per category) paid back with energy savings using current energy costs (June 2022) and preferential interest rates (5% APR). Further to the carbon footprint simulations, this information was compiled to fulfil ISO 14040:2006 certification. The certification file for the ISO 14040 standard does not provide additional inputs relevant for assessing the regeneration performance. However, the values considered for ISO certification can be used for parallel certification under the LEED and BREEAM schemes [32].

According to the simulation results, the regeneration strategy burden on the carbon balance of the neighbourhood is neutralized after two years facilitating, the carbon neutrality objective. Savings on energy consumption offer a positive balance for the investment after two years as well, which is an especially relevant outcome in light of the current evolution of energy costs.

The scores obtained for the neighbourhood after regeneration (Figure 8) show a very good result (silver certification) for LEED and an EXCELLENT certification in BREEAM for Benicalap. Both the LEED and BREEAM evaluations provide clear evidence of the effectiveness of the regeneration policy, allowing building certification levels for the neighbourhood to be attained, while prior to regeneration only 5% of the buildings in Benicalap (those constructed after 2007) reached certification level.

## 5. Discussion

The proposed methodology develops a new modular PCM panel to be used in a combined implementation strategy for neighbourhood regeneration strategies. The proposal verifies the technical verification and performance evaluation on efficient performance while incorporating the complete requirements supporting the regeneration process for best results as outlined by Cheng et al. [33].

Our simulation results are very clear on the economic balance. The proposed strategy can be implemented within a reasonable timeline with limited financial support, even considering the simulation accuracy.

Certification might be considered as a relevant quality strategy for regeneration, as the administrative processes involved favour a rigorous approach which present collateral advantages for cities. Certification adds a quality check for sharing city benchmarks among neighbourhoods and other cities.

Alternative applications of the proposed approach for Digital Twin-based services are relevant considering the continuous development of new Data Spaces; their combination can support a new urban sustainability assessment model [34].

## 6. Conclusions

The research presented on this paper has proposed a new sandwich solution for building envelopes. Starting from an innovative design, the newly developed ceramic PCM-embedded sandwich represents a modular and sustainable solution for neighbourhood regeneration, and has been technically characterized and evaluated for a whole neighbourhood.

Building upon careful selection of the base PCM material, the new modular sandwich solution has been fabricated and evaluated for energy performance. The ceramic foam support (with increased thermally active PCM surface) coated with a conductive graphite thin film for faster energy exchanges has demonstrated its excellent performance in laboratory tests, with a low production cost and estimated longer service times thanks to the use of a durable and cost-effective design.

Beyond the laboratory evaluation, the proposed sandwich solution has been virtually implemented in a neighbourhood-wide integrated strategy for urban regeneration. The simulation study case for the regeneration of the Benicalap neighbourhood in Valencia, Spain offers relevant evidence of the extended impact which can be obtained when combining efficient PCM solutions with Passive Haus ventilation and green infrastructure climate mitigation measures.

The simulation implemented through Digital Twin services has been proven as an excellent approach for evaluating the impact of urban regeneration strategies. Digital Twin simulations that have been adequately fine-tuned with better climate models provide more detailed energy assessments at the neighbourhood and building level, facilitating the evaluation of new building materials and strategies for future climate-neutral city regeneration.

Big Data repositories supporting Artificial Intelligence processes will help to incorporate machine-learning approaches for new Local Digital Twins. Therefore, better accuracy on the evaluation of new building materials will favour their market development and accelerate the uptake of net-zero cities.

For both cities and companies, developing a digital model for their neighbourhoods (or buildings) to measure the LCA of the desired objective is a very efficient way to begin sustainability assessment. There might be other marketing objectives linked to certification as well. Although certification obviously adds an external verification, which adds a marketing dimension, it may not be reasonable to target many standards simultaneously to address a wider market.

The GIS-based Digital Twin will be replaced soon with the mandatory advent of BIM models following new regulations in many EU countries. However, this approach will require the support of supporting Data Spaces for a more universal approach to the challenge of global climate change.

Evolving from the current standardization panorama based on ISO 14040:2006 to a comprehensive model for Sustainability Assessment at the urban level is required. Many standardization bodies are presenting new workgroups on digital twins for more focused action.

**Author Contributions:** J.O.-M. wrote the paper, prepared the research background, and analysed the experimental results. V.L.-M. developed the methodological approach and corresponding experimental work, and T.M.P. post-processed the obtained information to facilitate its analysis. All authors have read and agreed to the published version of the manuscript.

**Funding:** This research was co-funded by the European Commission through the H2020 project "Green Cities for Climate and Water Resilience, Sustainable Economic Growth, Healthy Citizens and Environments (GROW GREEN)" Grant Agreement: 730283.

**Institutional Review Board Statement:** Not applicable.

**Informed Consent Statement:** Not applicable.

**Data Availability Statement:** The data presented in this study are available on request from the corresponding author. The data are not publicly available at this moment as they will be published soon in a doctoral thesis.

**Conflicts of Interest:** The authors declare no conflict of interest.

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
