# Peer review of "City Regeneration through Modular Phase Change Materials (PCM) Envelopes for Climate Neutral Buildings"

_sustainability, doi:10.3390/su14148902_

Round 1

Reviewer 1 Report

The authors propose a modular solution to the problem of city regeneration and a detailed experimental study is carried out on this scheme. The paper is interesting and positive, particularly to other researchers working on city regeneration. The problem is well set out and the results are well-presented and discussed. 

1.       In the page 2 line 8,there is a clerical error, “Passivhaus” should be changed to “Passive haus”.

2.       In the page 8 line 20,there is a clerical error, “PCN panel” should be changed to “PCM panel”.

Author Response

Please kindly see attached PDF with our response.

Reviewer 2 Report

Dear authors,

Thank you very much for the possibility to read your paper and share my comments with you.

Please consider the following remarks and suggestions:

1)The literary review is too short and includes many links to internet sources. Review should definitely be expanded. More relevant references (original research articles) 2020-2022 could be added.

2)It is unclear what the word "this" means in the following sentences. It is recommended to add a noun connecting the previous and the current sentence after the word "this".

a) "This was obtained from a 2 am (average pore size) polyurethane foam soaked with a 60% solid content Standard Stoneware slurry (Euroatomizado), and then burned to 300°C (with reducing flame) for eliminating the polymer and leaving an extremely thin (conductive) black smoke layer for thermal conductivity enhancement (see microstructure obtained on figure 2)."

b) "This allows again a clear assessment on the effectiveness of the regeneration policy since it allowed to attain building certification levels for the neighbourhood while prior to the regeneration only 5% of the buildings (those constructed after 2007) reached certification level in the Benicalap case."

c) "This will allow the development of new building materials solutions for city regeneration [27]."

3) The Abstract requires revision. Thе Abstract should not be opening words but a summary of the main findings and results of the article. Shorten the introductory part and include brief formulations of new scientific results in the abstract. The Abstract should not refer the reader to the article; the Abstract should be read as an independent micro-article, emphasizing the research results.

4) The review part of the INTRODUCTION section is not like a search for a ready-made solution to the problem facing the author. The absence of such a solution in publications (research gap) has not been formulated. Consequently, the author's research is not properly motivated. The research object, the goal, and the study's objectives are not clearly defined at the end of the INTRODUCTION section.

5) "The proposed PCM sandwich [11] was fabricated over a stoneware scaffold". - It is unclear who offered the sandwich, this article's author, or Publication's authors [11].

6) The choice of the sandwich is not justified. Consequently, all the results obtained below are very local and are not of general scientific interest.

7) "Flat "green" tiles (1000 x 1000 mm with the desired thickness) of the scaffold are then fired to 1.000°C on an oxygen free atmosphere for sintering the scaffold with a pyro-lytic carbon layer reduced from the "black smoke" created during the greening stage." It is not clear what the words "green" and "black" mean. Whether it is the color of the materials, or it is some characteristics of the materials. In any case, the terminology of socio-political and popular science publications should be avoided.

8) The Köppen Climate Classification should be used in subsection 3.2.

9) The CONCLUSIONS section does not contain short statements of new scientific results obtained by the author. The academic evaluation is lacking. The results obtained by the author are valid for the small area of Benicalap. The author does not explain how these results can be used in another, even neighboring area. Accordingly, the value of the results obtained by the author for world science is not clear.

10) The CONCLUSIONS section contain many links on references. Conclusions must be original. It is not clear for what purpose authors used so many citations in this section.

11) The article is divided into two parts that are weakly connected. The first section is not a review part of a scientific article, as it does not motivate the need for research. The first section is more of an essay. The second part is a technical report on the study of a specific object without generalization and scientific understanding of the results obtained. 

Author Response

(The authors gave the same response as above.)

Reviewer 3 Report

1. The core of this paper is an introduction to the modular application of phase change materials in building envelopes, but this is not well reflected in the title. I think the current title does not focus on the core of the paper but is too broad.

2. How do phase change materials save energy? I believe this is of great interest to readers of this paper, but it is not clearly explained yet. Table 2 gives the energy efficiency of phase change materials at the level of the wall, how this is calculated at the building level is something I would suggest the authors explain further.

3.The conclusion section of the current paper is not written well enough. Normally, there are not so many references cited in the conclusion. The scholarly contribution of this paper is not well summarized in the current conclusions

Author Response

(The authors gave the same response as above.)

Round 2

Reviewer 2 Report

The authors took into account all the comments. But some remarks were taken into account a little formally. The review could still be expanded and the abstract more structured. But the paper could be published.

Author Response

Dear reviewer,

Thanks for the new review.

The abstract has been adapted for a better support of the proposed paper structure.

The literature review has been expanded with 3 new references providing wider scientific reference to the proposed research.

All the best,

Reviewer 3 Report

I believe that the current edition meets the basic requirements for publication.

Author Response

Dear reviwer,

Thanks for your review.

Best,